# What Drives the Spatial Heterogeneity of Urban Leisure Activity Participation? A Multisource Big Data-Based Metrics in Nanjing, China

Shaojun Liu [1,2,3,4], Xiawei Chen [2,3,4], Fengji Zhang [2,3,4], Yiyan Liu [5] and Junlian Ge [2,3,4,*]

1   School of Internet of Things, Nanjing University of Posts and Telecommunications, Nanjing 210023, China; lsj@njupt.edu.cn
2   Key Lab of Virtual Geographic Environment, Nanjing Normal University, Ministry of Education, Nanjing 210023, China; 211302133@njnu.edu.cn (X.C.); 211301023@njnu.edu.cn (F.Z.)
3   State Key Laboratory Cultivation Base of Geographical Environment Evolution, Nanjing 210023, China
4   Jiangsu Center for Collaborative Innovation in Geographical Information Resource Development and Application, Nanjing 210023, China
5   School of Geographic Information and Tourism, Chuzhou University, Chuzhou 239000, China; lauradew@163.com
*   Correspondence: gejunlian@njnu.edu.cn

**Abstract:** With the rapid pace of urbanization, enhancing the quality of life has become an urgent demand for the general public in both developed and developing countries. This study addresses the pressing need to understand the spatial distribution and underlying mechanisms of urban leisure activity participation. To achieve this, we propose a novel methodological framework that integrates diverse big data sources, including mobile phone signaling data, urban geospatial data, and web-crawled data. By applying this framework to the urban area of Nanjing, our study reveals both the temporal and spatial patterns of urban leisure activity participation in the city. Notably, leisure activity participation is significantly higher on weekends, with distinctive daily peaks. Moreover, we identify spatial heterogeneity in leisure activity participation across the study area. Leveraging the OLS regression model, we design and quantify a comprehensive set of 12 internal and external indicators to explore the formation mechanisms of leisure participation for different leisure activity types. Our findings offer valuable guidance for urban planners and policymakers to optimize the allocation of resources, enhance urban street environments, and develop leisure resources in a rational and inclusive manner. Ultimately, this study contributes to the ongoing efforts to improve the quality of urban life and foster vibrant and sustainable cities.

**Keywords:** urban sensing; leisure activity participation; correlation modeling; mobile phone signaling data; geospatial big data

## 1. Introduction

With the rapid pace of urbanization in recent decades, improving the quality of life (QOL) has become an imperative for both developed and developing countries [1–3]. Scholars contend that leisure, along with the resulting positive emotions and subjective well-being (SWB), plays a crucial role in determining one's overall quality of life [4–7]. Consequently, there is a need to measure the spatial distribution of urban leisure activity participation and understand its underlying mechanisms. This research offers valuable insights for evaluating urban space quality [8,9], residents' well-being [10], the rationality of service facilities [11], and urban vitality [12].

In the era of big data, the availability of ample samples and data-driven approaches provides reliable data and technical support for investigating human socio-economic activity [13–15]. Some studies employ user-generated content (UGC) big data, such as social media check-ins and web texts, to uncover the spatiotemporal dynamic and behavior

preferences of urban leisure tourism activities [16,17], or to predict potential leisure activity spaces within cities [18]. Others utilize remote sensing images, point of interest (POI) data, and spatial analysis techniques to examine urban leisure space [19] and nighttime leisure space [20]. However, the major shortcoming of the existing literature is the lack of a method to accurately estimate leisure activity participation at the city scale using unbiased samples, specifically for large populations.

A more intriguing topic is understanding the factors that drive participation in urban leisure spaces. Previous studies have attempted to correlate urban spatial design elements (such as road structure, facility allocation) and sociodemographic indicators with individual leisure preference and the distribution of leisure activities. For example, Liu et al. (2020) explored the spatial pattern of residents' daily leisure activities using the Beijing Official Household Travel Survey, analyzing the impacts of demographics (e.g., income level, the proportion of highly educated residents) and urban build environment characteristics (e.g., density of leisure venues, density of walkable street intersections) [21]. He et al. (2019) analyzed the relationship between street configurations and the density of leisure entertainment facilities, suggesting that indicators such as closeness, betweenness, severance, and efficiency can measure effective street network design [22]. However, there are several limitations in the literature: (1) the density of leisure facilities cannot fully capture the actual population participation in leisure activities, which serves as the dependent variable for driving modeling; (2) the differences in driving mechanisms for various types of activity have not been thoroughly discussed; (3) the measurement of spatiality impact indicators on leisure activity is incomplete, as density of leisure facilities and topologic features of road network structure only characterize a fraction of the physical environmental characteristics. Other factors, such as socio-economic features (e.g., consumption level), may also impact the spatial distribution of leisure activity.

To address these limitations, this paper proposes a novel methodological framework for measuring the spatial pattern of leisure activity participation (LAP) and explaining its driving mechanism by integrating multi-source big data, including mobile phone signaling (MPS) data, urban geospatial big data, and web clawed data. This framework allows us to answer two key questions: Where do people go for leisure? How does the spatial pattern of urban leisure activity participation emerge? By constructing individuals' daily activity chains using MPS data, we employ a machine learning method based on multiple spatiotemporal behavioral statistical features to infer leisure behavior and estimate the distribution of urban LAP. Furthermore, we design a quantitative index system to analyze the internal and external conditions of urban leisure regions (ULRs) and explain the reasons for the spatial heterogeneity in participation across various leisure activities. The findings of this study serve as a scientific reference for facilitating reasonable urban resource allocation, increasing opportunities for leisure activity participation, and promoting residents' well-being.

In this study, the urban area of Nanjing city is used as the experimental area, and one month of cell phone user activity chain data is collected. Fifteen spatiotemporal behavioral statistical indicators and four machine learning algorithms are employed to classify individual's activity chains. The random forest method, which achieves the highest accuracy, is selected for the estimation of LAP. By integrating the physical construction conditions and the surrounding economic and demographic characteristics of ULRs, multidimensional metrics are developed to analyze the mechanism of spatial heterogeneity of urban LAP. Using ordinary least squares (OLS) modeling, it is found that these indicators explain a high proportion of the variance for sports and tourism activities (85% and 67%, respectively), but a relatively lower proportion for cultural and recreational activities. The influence of each factor on the participation in different types of leisure activities varies. Internal physical construction conditions and external demographic and economic environment have a more pronounced impact on tourism and recreation activities, while sports and cultural activities are also influenced by subjective environmental perceptions. These findings provide important insights for leisure service planners and local governments in understanding the

spatial distribution patterns of urban leisure activity participation, the demand for various types of leisure activities, and people's expectation regarding the creation of internal and external environments for different leisure spaces.

The remainder of this paper is organized as follows. The following section describes the related works, the case study area, and the data used. Subsequently, we introduced the methods employed, including the estimation of leisure activity participation, quantification of driving indicators, and spatial correlation regression modeling. The subsequent sections present the results and discussion, followed by the conclusions.

## 2. Related Works

### 2.1. Quantifying Urban Leisure Space and Leisure Activity

Participation in leisure activities yields various benefits for different groups of individuals [23,24]. Given the accelerated urbanization, the emergence of high-density buildings, and the encroachment of public open space, the design of a well-planned neighborhoods has become increasingly crucial [25]. Additionally, providing a diverse range of leisure options is also essential [26]. Thus, it is necessary to quantitatively measure urban leisure activity space and people's participation in leisure activities to understand resource utilization and behavioral preferences. This knowledge can assist urban planners in developing practical design solutions and aid businesses in establishing scientifically informed site selection plans.

Currently, most relevant studies examine participation in urban leisure activities by focusing on subjective preferences, such as individuals' satisfaction with leisure experiences, the types of activities involved, and behavioral choices [27–29]. Furthermore, scholars have used POI data to analyze the spatial structure and clustering characteristics of various leisure activity places in cities, aiming to summarize the public activity space system of urban residents [19,30]. However, there is a lack of in-depth research examining the metrics of spatial participation across multiple types of urban leisure activities and their spatial characteristics.

Mobile phone data provide a reliable and real-time source of information, enabling automatic monitoring of individuals' calls and travel behavior [31,32]. Machine learning classification methods [33], statistical relationship learning techniques [34], and probabilistic graphical modeling [35] have been successfully applied to infer urban human activity types using mobile phone data. However, the existing literature lacks a methodological exploration of quantitative measures of people's participation in leisure activities. Particularly, the diversity of leisure activity types and the uncertainty associated with cell phone location data pose challenges for activity inference.

### 2.2. What Impacts Urban Leisure Activity Participation and Its Distribution?

Urban leisure behaviors are closely related to personal demographic characteristics and the physical environment [36]. Travel surveys and questionnaires have been extensively used to investigate the influence of demographic and built environment characteristics on leisure place selection [37,38]. Demographic factors, such as age, gender, educational level, and employment status, significantly affect decision regarding the types, duration, and locations of leisure activities [39–41]. For instance, households with medium and high annual incomes are less likely to engage in leisure activities within walking distance their homes [21]. Dargay and Clark (2012) emphasized that women travel less than men, the elderly less than younger individuals, and the employed and students more than other groups, as evidenced by National Travel Surveys [42]. Ambrey (2016) demonstrated that green urban environments and physical activity could enhance residents' well-being [37]. Research has indicated that spatial characteristics, such as the geographic accessibility of stores, food affordability, and the walkability of public open spaces, influence leisure participants [43,44]. However, survey data collection is time-consuming and labor-intensive, and may introduce errors due to respondent subjectivity.

The advent of big data and Geographic Information System (GIS) techniques presents new avenues for examining the geospatial effects on leisure activities. Studies have shown that access to green space and recreational facilities is closely associated with participation in leisure activities [19,45]. For instance, Cui et al. (2016) identified the distribution of population, transportation network, and commercial centers as key factors shaping the spatial pattern of karaoke venues, a significant urban leisure destination [46]. Liu et al. (2017) found that the policy, population, and economy factors dominate the spatial distribution of leisure venues [30]. He et al. (2019) explored the spatially stratified relationship between urban leisure entertainment activities and street configurations using spatial design network analysis [22]. They concluded that optimal street network design, characterized by metrics such as closeness, betweenness, severance, and efficiency, correlates spatially with the locations of leisure entertainment activities. These data-driven methods provide a more objective characterization of the urban environment and have enhanced our understanding of the factors impacting leisure activities.

Despite these advancements, only a limited number of studies have explored the effects of diverse factors on different leisure activities [47]. Until now, there is a lack of robust evidence on how various aspects, particularly geographic location, quantitatively impact urban leisure activity participation.

## 3. Study Area and Data Description

### 3.1. Study Area

According to the Annual report on China's leisure development (2019–2020) [48], there is a growing emphasis on leisure among Chinese people, resulting in an increase in their leisure time each year. Nanjing, the capital of Jiangsu Province and an important gateway city for the Yangtze River Delta region, has been chosen as the focus of this research. Renowned for its long history and status as a popular tourism destination, Nanjing's tourist and leisure venues, along with the main residential areas of its residents, are predominantly concentrated in the urban area (as depicted in Figure 1). Therefore, this article aims to explore the LAP and the driving mechanism behind leisure activities specifically within this urban area.

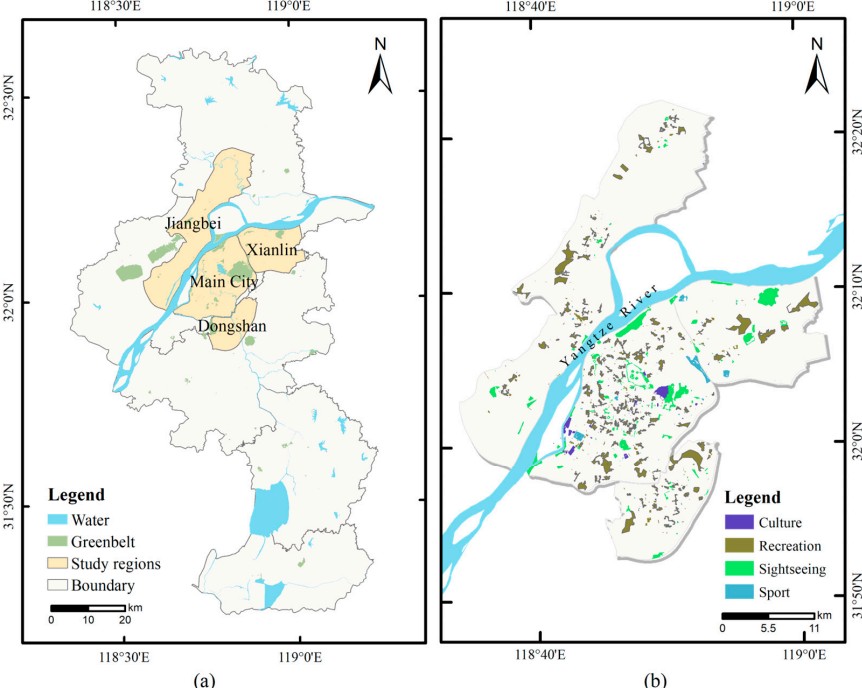

**Figure 1.** (**a**) is the case study area of Nanjing, China, and (**b**) shows the distribution of ULRs in our study area. The study area includes one main urban area and three sub-districts of Jiangbei, Xianlin, and Dongshan.

Defining the scope of leisure space is the basis for measuring urban LAP. In this study, Urban Leisure Regions (ULRs) are defined as accessible public space that offer relaxation, recreation, and entertainment services to people. By integrating the POI-based leisure clusters and Areas of Interest (AOIs), a total of 719 ULRs are identified, and the outcomes are illustrated in Figure 1.

The taxonomy of urban leisure venues has not yet formed a unified standard. This paper takes reference from the typology of leisure places in Wuhan, China, as proposed in [19], and categorizes urban leisure venues into four distinct types: culture, sports, recreation, and sightseeing. The type of ULRs within a given area also determines the specific leisure activities available in that area.

### 3.2. Data

For the estimation of LAP, we collected Mobile phone signaling (MPS) data from a communication operator, encompassing approximately 30 million subscribers for the month of April 2019. The MPS data records the geographical location and timestamps of various user activities such as making calls, sending and receiving short messages, switching base stations, accessing the mobile internet, and sending heartbeats to base stations (with a maximum interval of half an hour). This dataset is a reliable source for urban activity modeling and pattern mining [49,50].

To quantify the driving factors, we gathered leisure-related POIs, AOIs, urban roads, and building footprints from Gaode Maps, one of the most widely used web-map services providers in China. Additionally, we crawled Dianping POIs within the study area to evaluate the consumption level of each region. Dianping (http://dianping.com, accessed on 1 April 2020) is the first and most extensively used life services review website in China. Furthermore, we collected Baidu Maps Street View Image (BMSV) data to assess the subjective perception of street quality by the public. A total of 263,984 BMSV images we obtained for four grades of roads in the study area, with a sampling interval of 50 m. For each sampling point, this paper collected street view images from four angles: left, right, front, and back as shown in Figure 2.

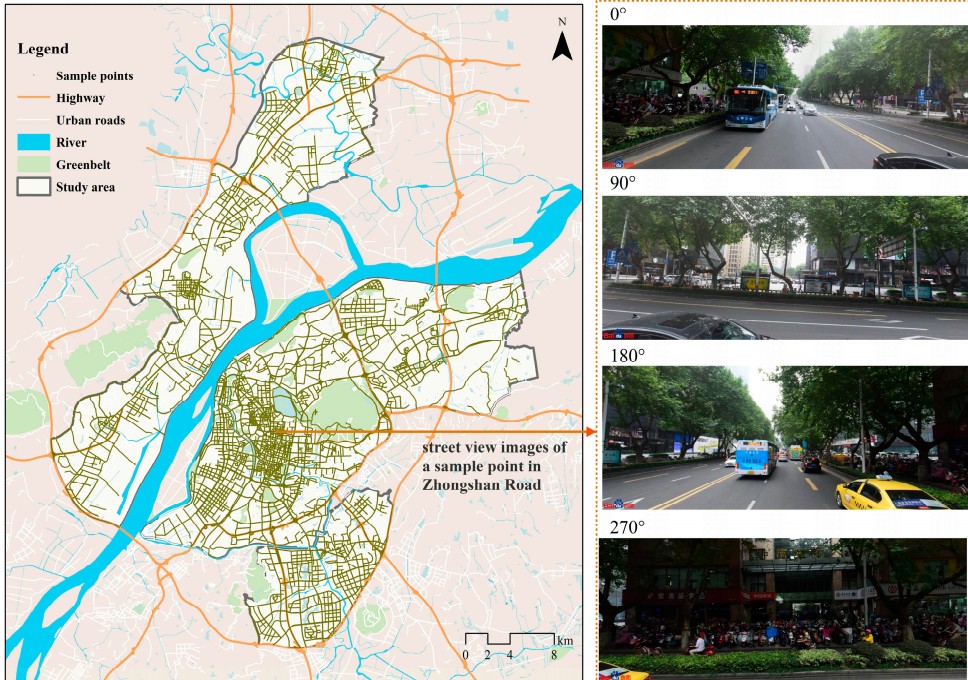

**Figure 2.** All collected points of BMSV images in the left figure and street map view image examples of four viewing angles.

The detailed information of all these datasets is provided in Table 1.

**Table 1.** Summary of experimental data information in this research.

| Data | Content | Usage [1] | Source | Time |
|------|---------|-----------|--------|------|
| MPS | 4.39 TB original MPS records | 1 | A Chinese telecommunications operator with a high subscriber market share | April 2019 |
| POI | 119,267 points' coordinates and their attributes | 2 | Gaode Maps, a Chinese map service platform (https://www.amap.com, accessed on 1 April 2020) | April 2020 |
| AOI | 9079 polygons with different functional attributes | 1, 2 | | |
| Buildings | Building profiles with a height attribute | 2 | | |
| Roads | 4 levels of urban road traffic network | 2 | | |
| Merchant POI | 22,659 merchant information, including average spending and ratings | 2 | Dianping website, China's most popular lifestyle service review site | |
| BMSV images | 65,996 street sampling spots along with the transportation network | 2 | Baidu Maps, another Chinese map service platform (https://www.map.baidu.com, accessed on 1 April 2020) | |

[1] The numbers in the third column represent: 1-LAP estimation; 2-evaluation of impact factors.

## 4. Methods

Based on the aforementioned experimental areas and datasets, this study presents a technical process (Figure 3) to accomplish two objectives: (1) quantitatively estimate the spatial–temporal pattern of urban leisure, and (2) construct a set of driving indicators to investigate the factors influencing this leisure pattern.

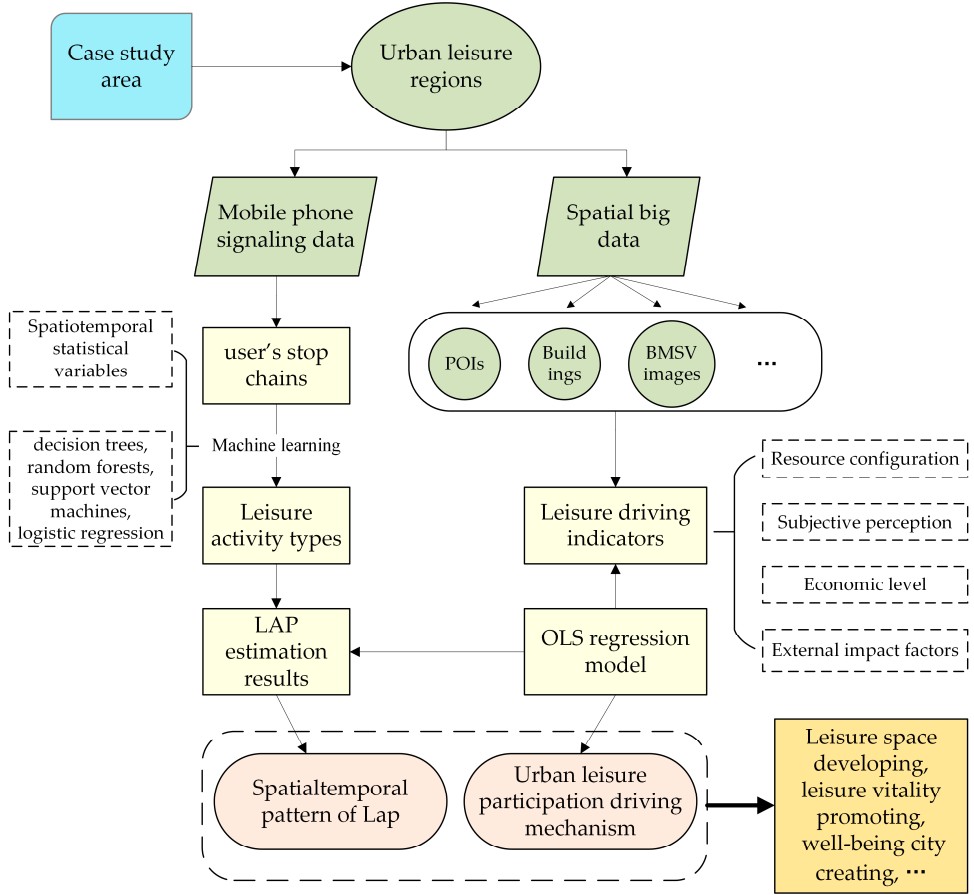

**Figure 3.** Framework for analysis of urban leisure participation patterns and their driving mechanisms.

### 4.1. Estimation of Urban Leisure Activity Participation Using MPS Data

Urban leisure activity participation (LAP) is introduced to capture the extent of people's involvement in different types of urban leisure activities. However, due to privacy concerns, the location data provided by phone operators often lack contextual information, resulting in a significant gap between the raw MPS data and the semantic interpretation of the trajectories. In order to bridge this gap, we first construct human stop chains based on the characteristics of MPS data and human daily behavior patterns. Subsequently, each stop is labeled using typical machine learning algorithms. Considering the uncertainty in the positioning of cellular base stations, we assign labels to the types of activities, including home, work, social visit, leisure, and others.

### 4.1.1. Construction of Stop Chains and Activity Types Labeling for Sample Data

To construct a user's activity trajectory (stop chains) over a month, this study follow three steps based on the location update mechanism and positioning accuracy of MPS data. Firstly, records with the same base station ID for each user are merged into a stay in chronological order. A visit is expressed by equation (1), which includes the base station ID, arrival time, departure time, and duration. Secondly, when the moving speed of two adjacent stay records is lower than the average walking speed of a human (4.96 ft/s [51]), the two stays are merged into a single visit. Finally, we correct the positional drift data caused by the signal oscillation (ping-pong effect), we obtain the user's stop chains as shown in Equation (2).

$$stay = \{id,\ arrive,\ depart,\ duration\}, \tag{1}$$

$$chain = \{(s_1, d_1), \ldots, (s_n, d_n)\}, \tag{2}$$

where $s_i$ is the $i$-th stay, and $d_i$ is the date of the $i$-th stop.

To obtain reliable training data, a random sample of 200 users' monthly stop chains is selected from the original dataset. It is observed that people generally follow fixed routines or scripts for their everyday rather than consciously planning them on a day-to-day basis. This generates high spatial stability and temporal periodicities in activity-travel behavior [52]. Therefore, several factors are considered for manually annotating the activity type of each stop in the 200 sampled users' stop chains. They include:

- The land use of the stop area;
- Time range of stop times (morning, afternoon, night, etc.) and the duration of each stop;
- Frequency of visiting different places;
- Whether it is a day of rest;
- Distance from the usual point of residence (home/place of work).

Based on the frequency, duration, and duration of stay, the most frequently visited place at night if often the user's home, while the most common daytime destination is typically their workplace. Additionally, activities near scenic spots, shopping malls, etc., are often associated with leisure activities, and longer durations of stay are expected on rest days.

### 4.1.2. Estimation of LAP Using Machine Learning Methods

Human activities are social behaviors that follow specific patterns in time and space [53,54]. Probabilistic modeling methods have been commonly used to infer urban activities based on land use characteristics [35,55]. However, knowledge-based approaches are typically employed for inferring the most common activities, such as work or home. While Graph Neural Network (GNN) models are effective in capturing high-dimensional features of data samples, determining optimal training parameters often leads to overfitting or underfitting. It is better to label the activity purpose of user stop chains by comparing multiple neural network models [56].

**(1)   Machine learning model and feature vector construction**

Building upon the study by Liu [56], we enhance the portrayal of spatial features by incorporating land-use feature vectors to improve the accuracy of activity type annotation. As shown in Table 2, we present 15 spatiotemporal statistical variables established to characterize people's activity behavior over time (one month in this study). These variables offer a systematic representation of behavioral patterns, including repetition, persistence, and diversity of activities, from a spatiotemporal statistics perspective. By using these spatiotemporal statistical features as input variables, a supervised learning method can effectively discriminate different activity types and provide a reliable basis for predicting the type of cell phone stop chains. Four commonly used machine learning methods, namely decision trees, random forests, support vector machines, and logistic regression, are selected to construct activity classification models. After training, the stay semantics of people are labeled as one of the following: home, work, social visit, leisure, or others. Stops with durations under 10 min are excluded from the identification target due to the location uncertainty of MPS data.

**Table 2.** Spatiotemporal statistical variables based on activity stay construction.

| Variable | Dimension | Description |
|---|---|---|
| Visit Frequency | Spatial | The frequency of visits to this stop point is divided by the frequency of visits to all stop points |
| Visit Frequency Week | | The frequency of visits to this stop point on a workday is divided by the frequency of visits to all locations |
| Land Use | | Land use structure of this stop point |
| Visit Frequency Weekend | | The frequency of visits to this stop on weekends is divided by the frequency of visits to all locations |
| Duration | Temporal | Dwell time of current stop point |
| Total Visit Duration | | The total time spent visiting the stop point is divided by the length of all dwell times |
| Earliest Visit Time | | The earliest appearance of the stop point |
| Latest Visit Time | | The most recent appearance of this dwell point |
| Average Visit Duration | | The average duration of visits to this dwell point |
| Variance Visit Duration | | The variance of the average duration of visits to this stop point |
| Longest Visit Duration | | Maximum time to visit the stop point |
| Total Visit Duration Week | | Total hours of visits to the stop on weekdays divided by the entire length of stay |
| Total Visit Duration Weekend | | Total hours of visits to the stop on weekends divided by the entire length of stay |
| Week | | 0-Workday, 1-Weekend |
| Day or Night | | 0-Night, 1-Day |

Since MPS data are non-precision positioning data, we use the 250-m and 500-m base station buffers as the reference plane for calculating land use characteristics in urban areas and suburban areas, respectively.

**(2)   Estimation of LAP based on spatial association**

In the subsequent step, the dataset of stop objects labeled as leisure activities is filtered, denoted as the set *L*. These labeled user stop objects are obtained using the machine learning methods by selecting the model with the highest training accuracy. Subsequently, candidate leisure regions are chosen for each stop object based on the spatial intersection between the stop region (representing the potential scope of human activity) and ULRs, which is noted as the set *U*. For the leisure activity object $l_i$, the area of all candidate ULRs intersecting the stop region are calculated and ranked from largest to smallest. Based on the ranking order of candidate objects, we assign the stop object to the corresponding ULR. The objective of this method is to establish the mapping relationship between each leisure activity object $l_i$ and ULR $u_j$, see Equations (3)–(6). Figure 4 illustrates some examples of the spatial association method. Finally, the urban LAP is derived by counting the number of stop activities associated with each ULR.

$$L = \{l_1, l_2 \ldots, l_i, \ldots, l_n\}, \tag{3}$$

$$U = \{u_1, u_2 \ldots, u_j, \ldots, u_m\}, \quad u = (uid, shape, type, ratio) \tag{4}$$

$$l_i = (stay, U), \tag{5}$$

$$f = (l_i \rightarrow u_j). \tag{6}$$

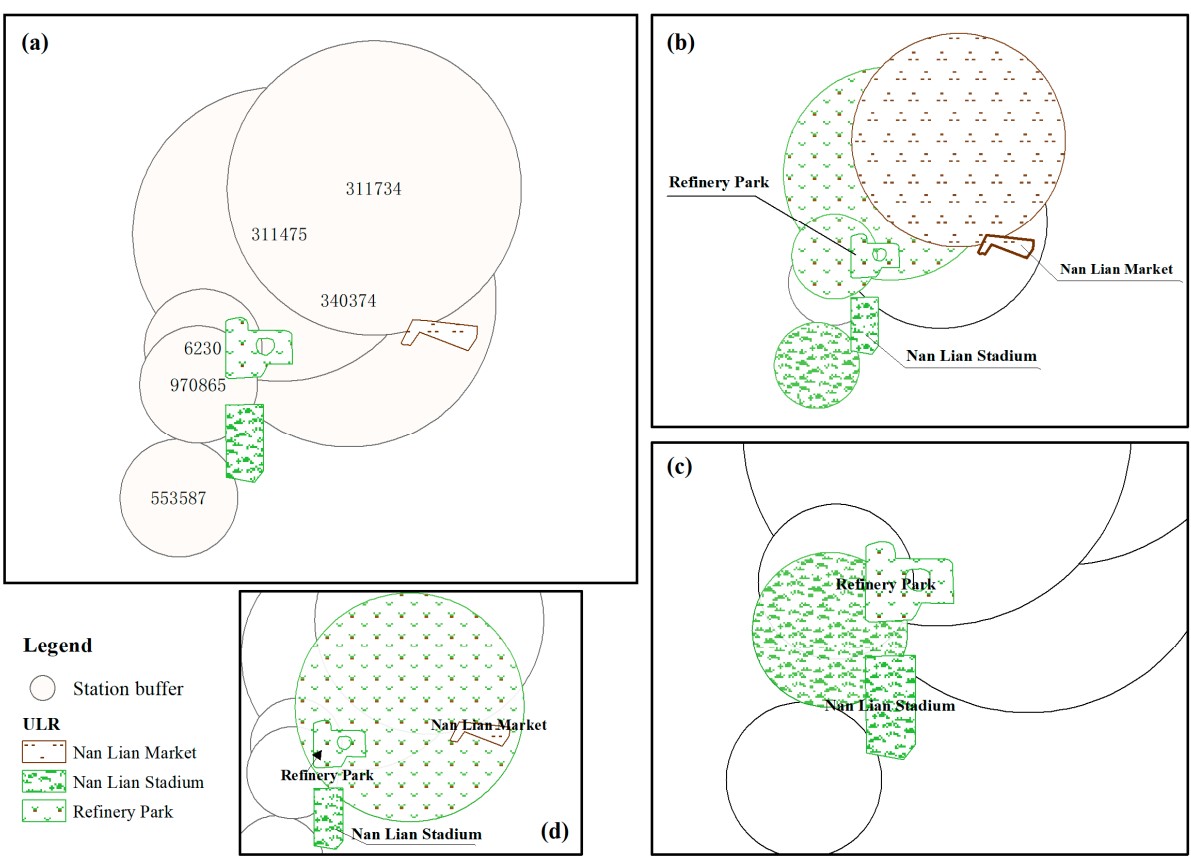

**Figure 4.** The example for the spatial association of ULRs and station buffers. (**a**) is the example stop regions and ULRs. (**b**) represents a stop region with only one ULR associated with it. (**c**,**d**) shows a stop region associated with multiple ULRs.

### 4.2. Construction and Quantification Driving Indicators System

From a spatial measurement perspective, this study constructs a comprehensive index system that influences the spatial distribution of urban LAP. The driving indicator assessment system consists of two dimensions: internal built conditions and external environment of ULRs. The internal conditions are quantified based on three aspects: physical construction, subjective perception, and consumer environment of ULRs. The external environment indicators aim to quantify the impact of the region on external connectivity and peripheral competition. Table 3 presents the driving factors, data sources, and calculation methods involved in the assessment.

**Table 3.** Indicators for the LAP driving analysis and their calculation methods.

| Evaluation Dimension | Evaluation Indicators | Data Sources | Quantitative Methods |
|---|---|---|---|
| Internal impact factors | **Resource configuration**<br>Density of leisure resources (Density)<br>Diversity of leisure resources (Diversity)<br><br>Richness of leisure resources (Richness) | Leisure POI | $Den = \frac{\sum_{i=1}^{n} Num_i}{Area}$<br>$Div = -\sum_{i=1}^{n} p_i \times \ln(p_i)$<br>$Ric = \sum_{i=1}^{n} p_i^0$ |

**Table 3.** *Cont.*

| Evaluation Dimension | Evaluation Indicators | Data Sources | Quantitative Methods |
|---|---|---|---|
| Internal impact factors | **Subjective perception** Greenness | Street Map View Images | $Gre = \frac{\sum_{i=1}^{n} vegetation_i}{\sum_{i=1}^{n} pixel_i}$ |
| | Openness | | $Ope = \frac{\sum_{i=1}^{n} sky_i}{\sum_{i=1}^{n} pixel_i}$ |
| | Walkability | | $Wal = \frac{\sum_{i=1}^{n} sidewalk_i}{\sum_{i=1}^{n} pixel_i}$ |
| | Enclosure | | $Enc = \frac{\sum_{i=1}^{n} building_i + pole_i + fence_i + trunk_i}{\sum_{i=1}^{n} pixel_i}$ |
| | **Economic level** Consumption level (CL) Consumption balance (CB) | Dianping POI | $Cl = Avg.loc[Itr(price)]$ $Cb = Std.loc[Itr(price)]$ |
| External impact factors | Traffic accessibility (TA) Surrounding population density (SPD) Homogeneous Competition Index (HCI) | Traffic network AOI, building outline POI | $Ta = \sum_{i=1}^{n} junction$ $Spd = \frac{\sum_{i=1}^{n} A_i \times h_i}{Avg \times Area}$ $Hcl = \frac{Num(ulr)}{min(distance)}$ |

$n$ in the first three formulas represents the number of POI types, $p_i$ is the number of POIs of $i$-th type; the fourth to seventh formula, the numerator is the sum of the number of pixels of the semantic segmentation objects, and the denominator is the total number of pixels of the street view images associated with each ULR; the eighth and ninth formulas denote the Inverse Distance Weight spatial interpolation function by $Itr()$, and $Std.loc[]$ is the standard deviation of the interpolation in each grid; in the tenth formula, $junction$ is the road intersections in a ULR; $n$ in the 11th formula represents the number of buildings, $h_i$ is the height of the $i$-th building, and $Avg$ is the floor area per capita (refers to China's urban per capita housing construction area of 31.8 m$^2$ in 2019); $Area$ in all formulas represents the area of ULR.

### 4.2.1. Evaluation of Internal Impact Factors

The quality of services provided by leisure places is the primary condition for urban leisure. Three indicators proposed by [57], namely density, diversity, and richness, are used to evaluate the spatial configuration of leisure resources. POI data related to leisure activities are used to assess the allocation conditions of resource. These POI types include dining service facilities, shopping service facilities, living service facilities, sports and leisure service facilities, accommodation service facilities, scenic spots, and scientific, educational, and cultural service facilities (including art museums, museums, exhibition halls, libraries, science and technology museums, etc.). Economic conditions form the basis for people's participation in leisure activities, as they are closely correlated with consumption and entertainment behaviors [30]. In this regard, we select two indicators, namely consumption level and consumption balance, to access urban economic characteristics using Dianping POI data.

The urban space plays a crucial role in promoting opportunities for relaxation and enhancing citizens' health [58]. A high-quality street space generally leads to positive subjective perception and can attract more people to participate in leisure activities [59]. To evaluate the quality of street space, four indicators are chosen: greenness (green space accessibility), openness, walkability, and enclosure. Dai et al. [60] concluded that greenness and openness contribute to an increased perception of beauty and decreased feelings depression. Walkability is associated with walk-based leisure activities [21]. Enclosure, which accounts for the sum of buildings, columns, and trunks, fosters a sense of safety among users [60], thereby providing more opportunities for physical activities [61].

To calculate the subjective perception indices of street landscapes, a generic semantic segmentation method called DeepLabV3 is applied to the image segmentation task. Over 500 street view images, including commercial centers, pedestrian streets, parks, etc., are annotated using the interactive segmentation tool EISeg [62]. The dataset is randomly split into 75% for training and 25% for testing. An adaptive learning rate is employed to train the model, and the related parameters are set as shown in Table 4. The pixel accuracy (PA) and intersection over union (IoU) are used to evaluate the performance of deep learning models. The trained model achieves a PA of 0.84 and an IoU of 0.62, indicating an acceptable

accuracy. Figure 5 showcases the semantic segmentation effect of street view images. Based on these results, the subjective perception index calculation method presented in Table 3 is utilized to evaluate the street landscape characteristics of ULRs.

**Table 4.** Parameter setting for semantic segmentation of street view images.

| Parameters | Values |
|---|---|
| Initial learning rate | 0.001 |
| Max-iter | 30,000 |
| Epoch | 200 |
| Batch | 4 |

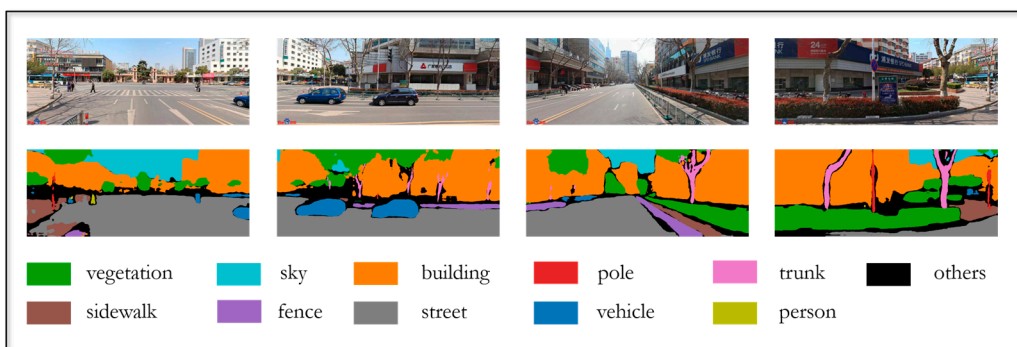

**Figure 5.** Examples of the semantic segmentation results for BMSV images.

### 4.2.2. Evaluation of External Impact Factors

The development of leisure areas and their spatial attraction are closely tied to favorable external conditions. Firstly, convenient transportation is a key factor influencing people's spatial choices and regional economic development [63]. The density of traffic network nodes is used an index to measure urban spatial traffic accessibility. Secondly, population distribution serves as the source of urban activities, and population density estimated from residential and office buildings is employed as an external driving indicator. Lastly, while the concentration of service facilities can attract human activities [64], spatial homogeneity also acts as a constraint that affects the maximum development benefits. The homogeneity competition index is used to measure the spatial competition of leisure areas by calculating the average Jaccard similarity coefficient between sets of leisure-related POI types in the target grid and those in the eight adjacent directions (East, West, South, North, Southeast, Northeast, Southwest, Northwest).

### 4.3. Spatial Correlation Regression Modeling of LAP

Ordinary least squares (OLS) regression [65] is utilized to estimate the relationship between leisure participation and driving indicators. To address the issue of multicollinearity, the Variance Inflation Factor (VIF) is employed to test for redundant variables in the model. If the VIF value of an explanatory variable exceeds 7.5, it indicates the presence of multicollinearity, and the variable is considered redundant. The results of OLS spatial regression encompass six aspects: model performance, coefficient of each explanatory variable, significance, steady-state, deviation, and spatial autocorrelation. Model performance is evaluated by calculating the Multiple R-Squared and Adjusted R-Squared indicators, which demonstrate the effectiveness of the constructed model in interpreting the dependent variable. The coefficients of the explanatory variables reflect the nature (positive or negative correlation) and strength of the relationship between them and the dependent variable. Joint F and chi-square statistics are employed to test the statistical significance of the model. A small probability value ($p$) returned by the statistical test indicates a low probability

of the coefficient being 0. If the possibility is less than 0.05, the coefficient is statistically significant with 95% confidence. The OLS regression equation is expressed as:

$$Y = \beta_0 + \beta_1 X_1 + \beta_2 X_2 + \ldots + \beta_k X_k + \varepsilon, \tag{7}$$

where $\beta_0 \ldots \beta_k$ is the regression coefficient, $X_1 \ldots X_k$ denotes the independent variable, and $\varepsilon$ is the random error. The principle of OLS regression is to find an optimal fitting curve that minimizes the sum of squared distances from each point to the straight line, resulting in the smallest sum of squared residuals: $RSS = \sum_{i=1}^{n} (Y_i - \hat{Y}_i)^2$.

## 5. Results and Discussion

### 5.1. What Is the Distribution Pattern of Urban Leisure Activity Participation in the Study Area?

By integrating remote sensing-interpreted land use data and urban AOIs, we obtained the current land use data of Nanjing, which consists of seven categories: arable land, commercial land, industrial land, educational land, public service land, residential land, tourism land, sports and leisure land, and water area. Using these land-use characteristics along with other spatiotemporal statistical variables as inputs, the accuracy of four machine learning models was evaluated using a randomly selected sample of 150 users' stay chains. The results, presented in Table 5. Table 5 indicate that the random forest model achieved the highest activity type inference accuracy of 92%. Accordingly, this model was chosen to label the type of users' stop chains in April, resulting in the successful identification of 2,890,267 leisure activities.

**Table 5.** Parameter settings and training accuracy for the activity type inference machine learning algorithms.

| Model | Title 2 | Title 3 |
|---|---|---|
| Decision trees | Criterion = gini<br>min samples split = 10<br>min samples leaf = 5<br>max depth = 90 | 0.904 |
| Random forest | n_estimators = 800<br>criterion = gini<br>max depth = 30<br>bootstrap = true<br>min samples split = 2<br>min samples leaf = 50 | 0.923 |
| Logistic Regression | solver = liblinear<br>penalty = l2<br>C = 1.0 | 0.665 |
| Support Vector Machine | default parameters | 0.709 |

Based on the aforementioned results of leisure activity identification, we analyzed the hourly variation of LAP and the frequency distribution of leisure duration. Figure 6 illustrates these findings, with half-hour and one-hour intervals used for statistical purpose. Figure 6a reveals a significant difference in the time distribution of leisure activity intensity between weekdays and weekends. Specifically, leisure activity participation is more prominent on weekends compared to weekdays. On weekdays, the first notable increase in leisure activity occurs between 8–9 am, while on weekends, this peak is delayed by one or two hours. The most active time for leisure activity on weekdays is between 5–8 pm, while on weekends, there are two peak hours, namely 10 am–3 pm and 5–7 pm, respectively. Figure 6b,c demonstrate that 90% of leisure activities last within 4 h on weekdays, and 90% do not exceed 4.5 h on weekends. This findings align with the common temporal pattern of human leisure activities [15,35]. Only a small proportion (about 8‰) of leisure stays extend beyond 10 h, which may be mistakenly identified as leisure activities when they are

actually represent secondary workplace locations or leisure areas near the residences of relatives and friends.

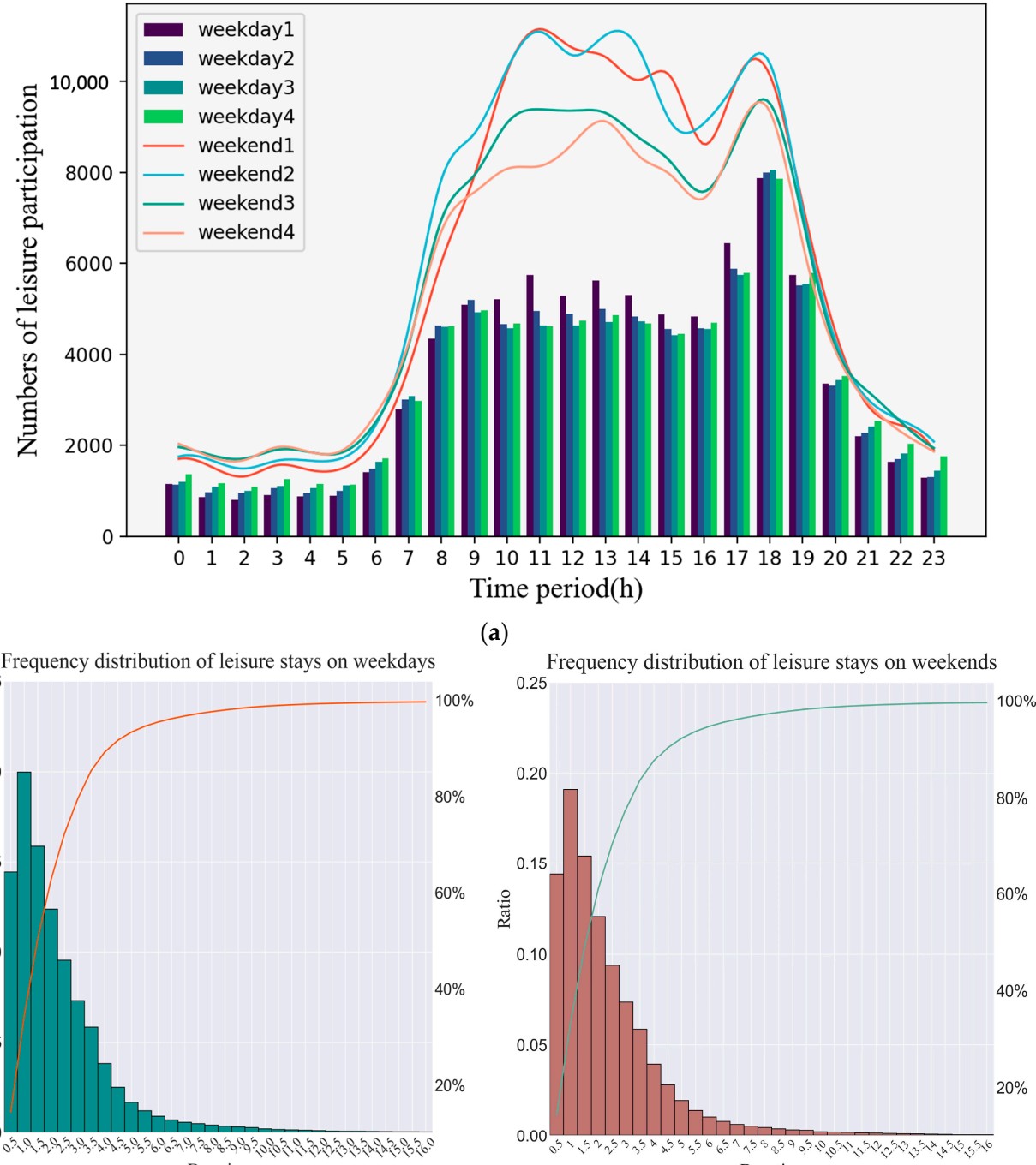

(**a**)

(**b**)
(**c**)

**Figure 6.** The temporal pattern of LAP during April in the study area of Nanjing. (**a**) shows the hourly changes of LAP on weekdays and weekends; (**b**,**c**) shows the frequency distribution of leisure duration on weekdays and weekends, the curves are the cumulative percentage change.

Finally, we examine the spatial distribution of LAP in the study area, as shown in Figure 7. High-intensity leisure activity areas are primarily concentrated in the central of main urban area and the southern sub-urban area. However, the Xianlin suburban area exhibits a lower intensity of leisure activities compared to other regions.

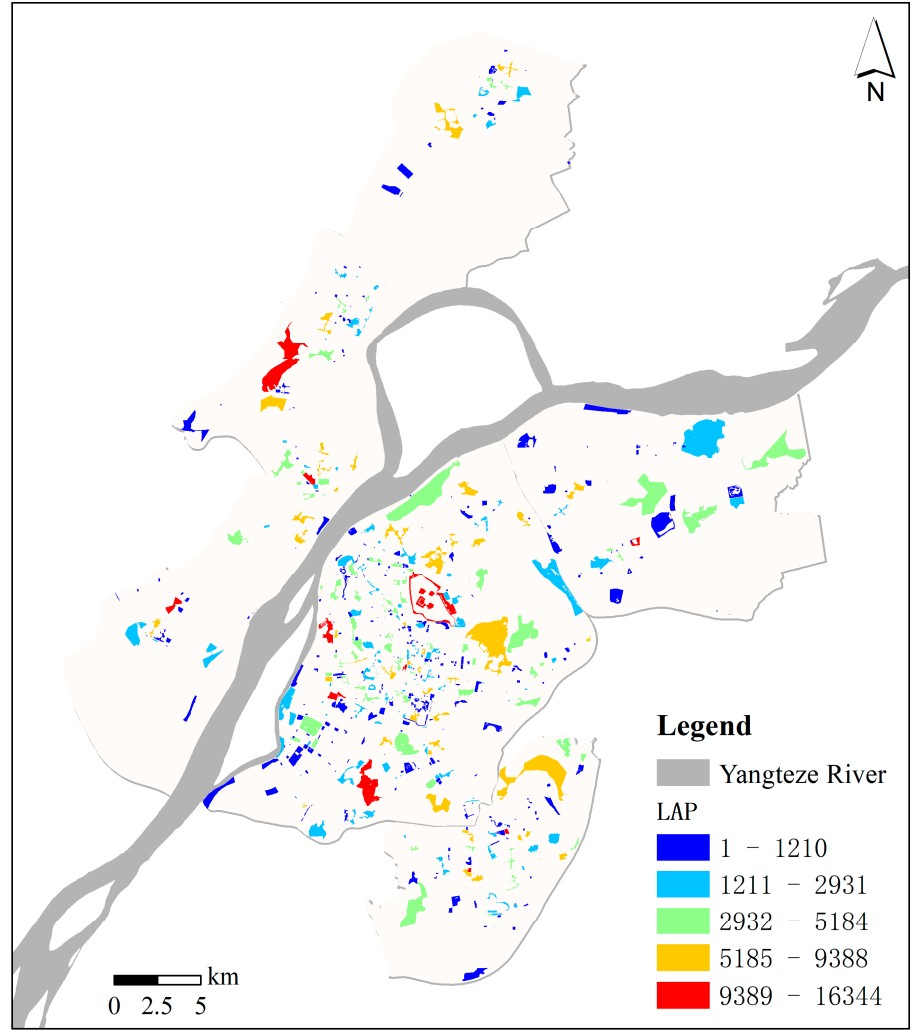

**Figure 7.** The spatial distribution of LAP during April in the study area of Nanjing.

The existing research on measuring urban leisure spaces has primarily focused on exploring the characteristics of the multi-center pattern of urban leisure spaces [19,30]. However, there has been limited investigation into the human activities within these leisure spaces. This study specifically examines the spatiotemporal patterns of activity participation in urban leisure spaces. We have identified regional imbalances in the development and utilization of current urban leisure resources, as well as temporal preferences in people's leisure choices. These findings provide essential decision-making foundations for optimizing local service resources and designing the leisure products.

*5.2. What Are the Impact Factors for the Distribution of LAP in the Study Area?*

According to Section 4.1, the rest days are the most active leisure time. In the subsequent analysis, we investigate the driving mechanism of urban leisure by LAP data from rest days as a proxy. The OLS spatial statistical model is employed to explore the associations between the LAP estimation results of rest days (Section 4.1) and the quantitative results of the driving indicators (Section 4.2). According to Figure 8, the histogram of the residuals demonstrates a close fit to the normal curve, indicating an unbiased model. Table 6 presents a summary of the correlation analysis. The *p*-value of the joint F-statistic and joint chi-squared statistic test confirm the statistical significance of the model. The VIF values indicate the absence of redundancy among the explanatory variables, suggesting no strong multicollinearity among the LAP driving indicators. The model achieves an R-squared value of 0.5234, indicating an explanatory power of 52.3% for urban leisure

activities. Among all explanatory variables, five indicators are statistically significant (*p*-value < 0.05): density of leisure resources, diversity of leisure resources, richness of leisure resources, traffic accessibility, and surrounding population density.

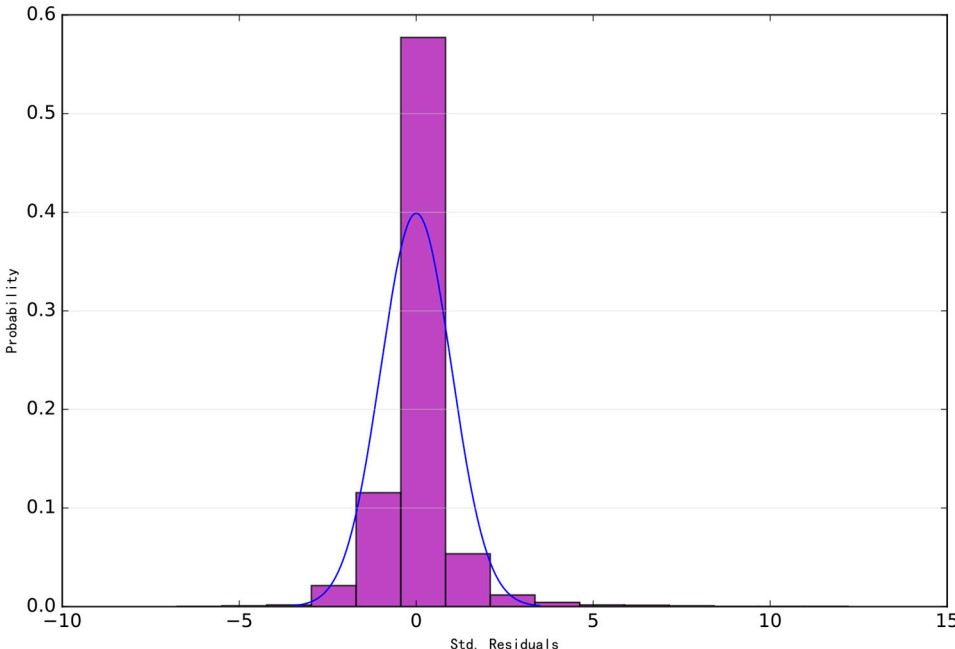

**Figure 8.** Histogram of standardized residuals.

**Table 6.** Results of the OLS modeling for the LAP driving mechanism exploration.

| Diagnostics of OLS Modeling | |
|---|---|
| Multiple *R*-Squared | 0.5234 |
| Adjusted *R*-Squared | 0.5106 |
| Joint F-Statistic | 41.003 *** |
| Joint chi-squared Statistic | 367.354 *** |
| Jarque-Bera Static | 524.155 *** |

| | | | | | Summary of each explanatory variable | | | |
|---|---|---|---|---|---|---|---|---|
| Evaluation dimension | Indicator | Min | Max | Mean | Standard deviations | Coefficient | Probability | VIF |
| | **Independent variable** | 1 | 15,450 | 2160.540 | 2422.495 | \ | \ | \ |
| Internal | Density of leisure resources | 0 | 3.523 | 0.228 | 0.387 | −588.290 | 0.013 ** | 1.355 |
| | Diversity of leisure resources | 0 | 3.525 | 1.821 | 1.243 | −347.177 | 0.001 *** | 3.293 |
| | Richness of leisure resources | 0 | 83 | 21.355 | 21.169 | 93.962 | 0.000 *** | 5.263 |
| | Consumption level | 9.378 | 525.687 | 69.016 | 53.701 | 1.921 | 0.427 | 2.703 |
| | Consumption balance | 0 | 438.817 | 25.214 | 39.887 | −2.372 | 0.469 | 2.730 |
| | Greenness | 0.004 | 0.734 | 0.263 | 0.117 | 73.629 | 0.946 | 2.668 |
| | Openness | 0.001 | 0.513 | 0.112 | 0.072 | 346.093 | 0.833 | 2.247 |
| | Walkability | 0 | 0.124 | 0.018 | 0.015 | −3611.870 | 0.515 | 1.161 |
| | Enclosure | 0.006 | 0.637 | 0.190 | 0.117 | 218.155 | 0.170 | 3.613 |
| External | Traffic accessibility | 3 | 795 | 161.557 | 102.775 | 2.285 | 0.012 ** | 1.405 |
| | Surrounding population density | 0 | 55,760 | 2774.141 | 6503.236 | 0.029 | 0.051 * | 1.511 |
| | Homogeneous Competition Index | 0 | 30.723 | 0.102 | 1.446 | −18.789 | 0.742 | 1.095 |

Notes: ***, ** and * represents significance level of 0.1%, 1% and 5%, respectively.

Obviously, LAP exhibits greater sensitivity to the impact of objective construction conditions and changes in external physical factors. However, the subjective dimension does not exert an overall impact on leisure activity. Additionally, high population density, enhanced accessibility, and abundant service facilities demonstrate a positive driving effect on participation in leisure activities, without necessarily requiring a large number of densely distributed service facility resources.

It is important to note that different activities elicit distinct motivations and entail varying requirements for resource allocation and participation experience in different activity areas. Attempting to explain all leisure activities in the city using a single regression model would be challenging, as various leisure activities are driven by different mechanisms. Therefore, in the subsequent section, we will explore the characteristics of leisure activity participation driven by multiple factors for the four types of leisure activities.

*5.3. What Shapes the Distribution of LAP for Various Types of Leisure Activities?*

We developed regression analysis models to examine the driving mechanisms behind four types of leisure activities: sports, sightseeing, culture, and recreation. The results, presented in Table 7 (empty cells indicate factors with multicollinearity that were excluded), indicate the performance of the models. The sports model achieved the highest R-Squared value of 0.8571. The regression analysis model for the other three types of leisure activities achieved R-squared values of 0.6735 for sightseeing, 0.4912 for recreation, and 0.4665 for culture.

**Table 7.** Results of the OLS modeling for the LAP driving mechanism exploration.

| Indicator | Summary of Sports Activity Regression Analysis | | | Summary of Sightseeing Activity Regression Analysis | | | Summary of Cultural Activity Regression Analysis | | | Summary of Recreation Activity Regression Analysis | | |
|---|---|---|---|---|---|---|---|---|---|---|---|---|
| | Coefficient | Probability | VIF | Coefficient | Probability | VIF | Coefficient | Probability | VIF | Coefficient | Probability | VIF |
| Density | −9489.563 | 0.234 | 2.231 | −672.657 | 0.147 | 1.194 | −6674.508 | 0.063 | 2.665 | −496.207 | 0.096 | 1.317 |
| Diversity | 796.860 | 0.023 * | 3.580 | −703.496 | 0.000 *** | 3.450 | 576.951 | 0.098 | 7.891 | −549.905 | 0.010 ** | 4.529 |
| Richness | | | | 176.350 | 0.000 *** | 6.249 | 43.073 | 0.562 | 8.492 | 105.178 | 0.000 *** | 4.906 |
| CL | | | | −0.585 | 0.798 | 1.220 | −5.056 | 0.040 * | 1.934 | 4.657 | 0.238 | 3.860 |
| CB | | | | −7.497 | 0.245 | 1.365 | 12.572 | 0.069 | 2.542 | −4.686 | 0.329 | 3.804 |
| Greenness | 9822.498 | 0.038 * | 8.674 | 170.535 | 0.860 | 2.084 | −1790.725 | 0.059 | 1.758 | −174.452 | 0.927 | 2.905 |
| Openness | 7287.396 | 0.211 | 3.482 | 1112.127 | 0.438 | 1.869 | −4659.151 | 0.033 * | 1.782 | 204.247 | 0.942 | 2.767 |
| Walkability | 24,708.954 | 0.151 | 1.946 | 631.312 | 0.905 | 1.373 | 8896.126 | 0.436 | 2.270 | −9955.606 | 0.276 | 1.109 |
| Enclosure | 14,401.253 | 0.049 * | 9.328 | −505.827 | 0.679 | 2.236 | | | | 1317.330 | 0.520 | 4.155 |
| TA | 2.989 | 0.221 | 3.278 | 4.604 | 0.000 *** | 1.650 | −0.875 | 0.727 | 2.237 | −0.011 | 0.993 | 1.607 |
| SPD | −0.915 | 0.285 | 1.792 | −0.105 | 0.478 | 2.508 | −0.771 | 0.183 | 1.108 | 0.050 | 0.005 ** | 1.505 |
| HCI | | | | −45.982 | 0.196 | 1.271 | −4652.870 | 0.106 | 1.554 | 4646.075 | 0.014 * | 1.043 |

Notes: ***, ** and * represents significance level of 0.1%, 1% and 5%, respectively.

For sports, the diversity of POIs, street greenery, and enclosure significantly influence participation in this activity. Usually, sports activities include many outdoor programs, and the pleasant subjective experience created by street design plays a crucial role in motivating people to engage in these activities. Sports venues should be well-landscaped and offer a wide range of facilities to provide participants with various choices (according to Enclosure and Diversity indicators).

For sightseeing, transportation convenience and the availability of service facilities emerged as the most important influential factors on participation. Diverse leisure service resources, rather than balanced ones, have a greater impact on stimulating tourism activities, as indicated by the coefficients of Diversity and Richness. However, the streetscape measured by BMSV does not directly reflect the subjective feelings associated with the internal environment of the scenic area, resulting in an insignificant correlation between streetscape-related indicators and participation in tourism activities.

Cultural activities exhibit greater attractiveness due to the events held at individual venues, making it challenging explain the driving mechanism and resulting in the lowest R-squared value in the regression analysis. The results of the OLS modeling suggest that the openness of the activity area and the consumption level have a more pronounced impact. Cultural leisure participation is typically lower in developing countries, which

is confirmed by the negative relation between consumption level and cultural leisure participation. Additionally, the presence of abundant vegetation and buildings surrounding cultural venues create an enclosed environment that enhances the sense of history and cultural heritage. Figure 9 illustrates the street appearance of two popular cultural venues in Nanjing.

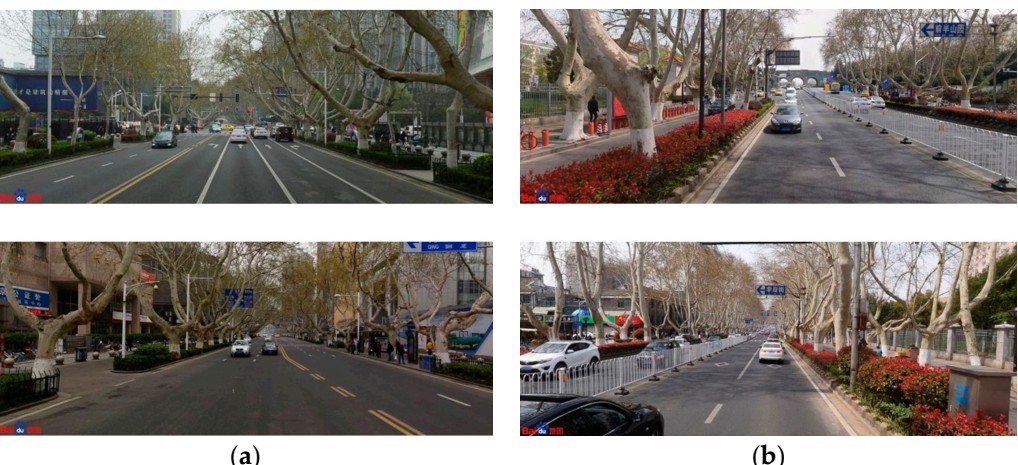

(**a**)          (**b**)

**Figure 9.** Streetscape images of two attractive cultural venues in Nanjing (front and rear view). (**a**) is the street scene near Nanjing Culture and Art Center and (**b**) is the street scene near Nanjing Museum.

For recreation activities, both internal and external environmental factors significantly influence LAP. Similar to sightseeing activities, recreation area require rich service facilities, and the need for balance is not prominent. Moreover, spatial location plays a crucial role in determining leisure participation, requiring a large resident population and a cluster of recreation areas in the vicinity. The HCI indicator shows a positive correlation with recreational activity participation and a negative correlation with other leisure activities.

Current research on the motivation and influencing factors of residents' leisure activities primarily focuses on establishing the correlation between demographic characteristics and leisure activity participation [39–42]. Some studies have indicated that street network structure and proximity to commercial centers significantly influence urban leisure and recreational activities [19,22,45,46]. In this study, we have developed a multidimensional and comprehensive indicator system to quantify the associations between various factors and different leisure activities. We have found that:

1. The identified factors have demonstrated a significant level of explanatory power for sports and sightseeing activities, reaching 85% and 67%, respectively. However, measuring the influence mechanism of recreational activities poses more challenges.
2. The resource conditions of ULRs are associated with sports, sightseeing, and recreation activities. Sports activities require a balanced allocation of service resources, whereas tourism and recreation emphasize the abundance of resources.
3. Participation in sports and cultural activities is influenced by the subjectively perceived of the environment created within the recreational area.
4. The LAP of recreation activities is strongly linked to the density of the surrounding population and the concentrated distribution of recreation and leisure areas.

Through our experiments, we have identified the correlations between the degree of participation in urban leisure activities and factors such as the richness and diversity of leisure venues, population density in the surrounding area, spatial accessibility, and competition from similar resources. Importantly, these relationships exhibit noticeable variations across different types of leisure activities. These findings provide valuable guidance for the context-specific development of leisure resources, reducing homogeneous competition, and enhancing the design and aesthetics of street environments.

*5.4. Limitations*

This study focus on a single city as an example, and there is scope for improvement in quantifying certain factors. Future research will address the following aspects:

- By incorporating other cities for comparisons, we will analyze the similarities and differences of urban LAP across different cities and evaluate the adaptability of the proposed indicator systems.
- The evaluation of subjective perceptual characteristics of indoor and outdoor environments within each leisure area (subjective perception assessment based on real pictures) will be integrated into the analysis system to further enhance the understanding of leisure-driven mechanisms.

## 6. Conclusions

Leisure activities play a significant role in subjective well-being. More and more scholars have focused on the important value of urban leisure activity participation metrics and their impact mechanism analysis. In this study, we propose a method to estimate urban LAP for each ULR by integrating multi-source geospatial big data and human activity tracking data. Using Nanjing as a case study, we explore the driving mechanism behind LAP distribution. The leisure and sightseeing activities of people often vary seasonally. Therefore, our findings primarily reflect the leisure participation characteristics during the spring season in the study area.

Our analysis reveals that LAP is significantly higher on weekends compared to weekdays, with two peaks in a day. The results highlight the regional imbalances in the development and utilization of urban leisure resources, as well as temporal preferences in peoples' leisure choices. The identified factors have achieved a significant level of explanatory power, particularly for sports and sightseeing activities, with 85% and 67% accuracy, respectively. We have also identified the specific driving factors for each type of leisure activity, such as street design for sports, transportation convenience for sightseeing, and cultural heritage for cultural activities. Additionally, we have observed the influence of resource conditions and population density on leisure participation. These findings offer valuable insights for optimizing local service resources, enhancing the design of leisure areas, and improving the aesthetics of street environments.

While this study provides significant contributions, there are some limitations. The research is based on a single city, and future studies should include multiple cities to compare and evaluate the adaptability of the proposed indicator systems. Additionally, further research incorporating subjective perception assessments of indoor and outdoor environments within leisure areas will enhance our understanding of the mechanisms driving leisure activities. By addressing these limitations, we can gain a deeper understanding of urban LAP across different contexts and improve our analysis system.

**Author Contributions:** Conceptualization, Shaojun Liu and Junlian Ge; methodology, Shaojun Liu and Xiawei Chen; software, Yiyan Liu; validation, Shaojun Liu, Xiawei Chen and Fengji Zhang; formal analysis, Shaojun Liu; investigation, Shaojun Liu; resources, Shaojun Liu and Fengji Zhang; data curation, Shaojun Liu and Fengji Zhang; writing—original draft preparation, Shaojun Liu; writing—review and editing, Shaojun Liu, Xiawei Chen, Junlian Ge and Fengji Zhang; visualization, Shaojun Liu; supervision, Shaojun Liu; project administration, Shaojun Liu, Junlian Ge; funding acquisition, Shaojun Liu. All authors have read and agreed to the published version of the manuscript.

**Funding:** This research was funded by National Natural Science Foundation of China, grant number 42301528 and Open Fund of Key Laboratory of Virtual Geographic Environment, Ministry of Education, grant number 2023VGE01.

**Data Availability Statement:** The data presented in this study are available on request from the corresponding author. The data are not publicly available due to the need for privacy protection of the user's location.

**Acknowledgments:** The experimental data used for this paper are supported by Jiangsu intelligent insight big data center of China telecom Co., Ltd. (Nanjing, China).

**Conflicts of Interest:** The authors declare no conflict of interest.

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
