# Peer review of "What Drives the Spatial Heterogeneity of Urban Leisure Activity Participation? A Multisource Big Data-Based Metrics in Nanjing, China"

_ijgi, doi:10.3390/ijgi12120499_

Round 1
Reviewer 1 Report
Comments and Suggestions for Authors
The format of the literature citation is not starndard. For example, in line 50, "[21] used the Beijing Official ......". In line 136 "Dargay & Clark [41] 2012) Highlighted ......" In line 176 "This paper refers to [19] typology of ......"etc. Please check the whole manuscript and rectify the literature citation format.
In line 287 to 288, what does the content mean "the text following an equation need not to be a new paragraph..."
Descriptive statistics of the dependent and independent variables should be added in the manuscript. Including mean, max, min and standard deviations.
In line 427 to 428, how do you judge that the residuals of OLS matches the normal distribution? You could use PP Plot , Q-Q plot or K-S test to verify your conclusion.
Comments on the Quality of English LanguageExtensive editing of English language required.
Author Response
Many thanks for your valuable time and constructive comments on our manuscript. Changes in the initial version of the manuscript are highlighted within the document using the track changes mode in MS Word. Below, we provide a point-by-point response to your comments.
- “The format of the literature citation is not starndard. For example, in line 50, "[21] used the Beijing Official ......". In line 136 "Dargay & Clark [41] 2012) Highlighted ......" In line 176 "This paper refers to [19] typology of ......"etc. Please check the whole manuscript and rectify the literature citation format.”
Response: Thanks for your comments. We have checked all the manuscript and corrected all the ciation format, as demonstrated in lines 12, 16-17 on page 2, lines 46, 48 on page 3, lines 6, 8, 10 on page 4, lines 29, 31 on page 8, and lines 1, 3 on page 12.
- “In line 287 to 288, what does the content mean "the text following an equation need not to be a new paragraph...".”
Response: Thank you for raising this question. This is unnecessary content in the document template, and we have already removed that paragraph (page 10).
- “Descriptive statistics of the dependent and independent variables should be added in the manuscript. Including mean, max, min and standard deviations.”
Response: Thank you for your suggestion. We have added the descriptive statistics of the dependent and independent variables in Table 6 on page 16.
- “In line 427 to 428, how do you judge that the residuals of OLS matches the normal distribution? You could use PP Plot , Q-Q plot or K-S test to verify your conclusion.”
Response: Thank you for your improvement suggestion, we have added the figure of residuals distribution in Figure 8 on page 17.
- “Extensive editing of English language required.”
Response: We apologize for the poor language quality of our manuscript. After conducting a thorough review of the article, we addressed the readability issues you pointed out. Additionally, we have enlisted the assistance of a native English speaker to meticulously check the language and assist us in polishing it. We sincerely hope that the flow and language level of the revised manuscript have been substantially improved.
Reviewer 2 Report
Comments and Suggestions for Authors
There is a growing interest in recreational sports and health, and I think this topic is very relevant. However, the following issues need to be addressed to meet the criteria for publication.
1. I think leisure activity participation is seasonally characterized. The authors used data from April, and I think the findings of this paper apply more to people's leisure activity participation in the spring than to other seasons.
2. I don't think it's appropriate to use the serial number of a reference as the subject of a sentence.
3. Acronyms should be explained the first time they appear, for example "URLs" on page 2, POI on page 1, AOI on page 4……
4. Authors are requested to proofread the manuscript carefully, as the references to figures and tables in the text are repeatedly incorrectly displayed as "Error! Reference source not found".
5. On line 411, the serial number of the figure is incorrect.
6. The article has a discussion section, but it seems to lack in-depth discussion.
It is suggested to improve the discussion section. In this section, the authors should present some kind of comparison of the results and methods of the present study with those obtained in previous studies. How similar or different are the findings and methods, and what new insights have been gained should also be highlighted.
7. In the section of conclusions, the authors should build on the correction suggested for the section of discussion by highlighting the significance as well as implications of the methods and results and not just a repetition of the findings as currently done.
Comments on the Quality of English Language
Minor editing of English language required
Author Response
Many thanks for your valuable time and constructive comments on our manuscript. Changes in the initial version of the manuscript are highlighted within the document using the track changes mode in MS Word. Below, we provide a point-by-point response to your comments.
- “I think leisure activity participation is seasonally characterized. The authors used data from April, and I think the findings of this paper apply more to people's leisure activity participation in the spring than to other seasons.”
Response: Thank you for your suggestion. We have incorporated the viewpoint you raised in the conclusion section of this article, as seen in lines 2-4 on page 20.
- “I don't think it's appropriate to use the serial number of a reference as the subject of a sentence.”
Response: Thanks for your careful inspection. We have thoroughly checked the manuscript and corrected all the ciation format, as demonstrated in lines 12, 16-17 on page 2, lines 46, 48 on page 3, lines 6, 8, 10 on page 4, and lines 29, 31 on page 8.
- “Acronyms should be explained the first time they appear, for example "URLs" on page 2, POI on page 1, AOI on page 4……”
Response: Certainly. We have included the full spelling of proprietary abbreviations for every first occurrence in the text, such as subjective well-being (SWB) and point of interest (POI). Furthermore, we have provided explanations for the concepts introduced in this article. For instance, the meaning of urban leisure region (ULR) is elaborated upon in lines 32-34 on page 4.
- “Authors are requested to proofread the manuscript carefully, as the references to figures and tables in the text are repeatedly incorrectly displayed as "Error! Reference source not found.”
Response: I apologize for the formatting issue. This could be an error caused by the version of the Word software. We have thoroughly reviewed the entire document and corrected these formatting errors.
- “On line 411, the serial number of the figure is incorrect.”
Response: Thank you for pointing this out. We have corrected the serial number of this figure, as demonstrated in line 1 on page 15.
- “The article has a discussion section, but it seems to lack in-depth discussion.
It is suggested to improve the discussion section. In this section, the authors should present some kind of comparison of the results and methods of the present study with those obtained in previous studies. How similar or different are the findings and methods, and what new insights have been gained should also be highlighted.”
Response: We are grateful for your suggestions. In the final paragraph of section 5.1 and the concluding paragraphs of section 5.3, we have incorporated a comparative analysis between the findings of our research and relevant studies (lines 8-15 on page 15 and lines 11-34 on page 19). We emphasize the novel discoveries brought about by our proposed research perspective, highlighting both its methodological value and its practical application in guiding urban planning practices.
- “In the section of conclusions, the authors should build on the correction suggested for the section of discussion by highlighting the significance as well as implications of the methods and results and not just a repetition of the findings as currently done.”
Response: Thanks for giving us so valuable advice. Combined with the modifications made to the discussion section, we have rewritten the conclusion of the paper. We summarize the most significant findings of this study, emphasizing the importance, value, and impact of the method proposed. Additionally, we highlight future directions for further research.
- “Minor editing of English language required.”
Response: We apologize for the poor language quality of our manuscript. After conducting a thorough review of the article, we addressed the readability issues you pointed out. Additionally, we have enlisted the assistance of a native English speaker to meticulously check the language and assist us in polishing it. We sincerely hope that the flow and language level of the revised manuscript have been substantially improved.
Reviewer 3 Report
Comments and Suggestions for Authors
In writing the abstract, I suggest that authors follow the prescribed instructions and add a background: Place the issue being addressed in a broad context and highlight the purpose of the study. Also, clarify the part that refers to the conclusion, as the last sentence is too general in this form.
The introduction clearly introduces the reader to the topic, citing the relevant literature and the uniqueness of this work compared to existing works. I suggest briefly describing the structure of the paper at the end of the introduction to give the reader an insight into the text of the following paper.
Related works
Lines 105-112: Since the chapter refers to previous work, the facts mentioned should be supported by literature. If it is the authors' opinion, it should be moved to the discussion chapter.
Line 113 starts with the term "Related research mostly starts..." and it is important to indicate which literature this part of the text refers to. This way it is easier to review and gain insight into the topic being addressed. Within the sentence, the authors only give the reference [28], and what is written suggests more than one literature.
The sentence in line 118 begins with the text "Few studies...", and again it is necessary to indicate which literature the facts mentioned refer to.
The literature citation in line 136 contains an error: "Dargay & Clark [41]2012)"
Due to the extensive nature of the paper, it is necessary to depict the research process with a flowchart.
"Error!Reference source not found." appears in many places in the paper, e.g., in line 258
Lines 287-88 should be deleted.
Line 414, please check if "8‰ of leisure duration" is written correctly.
Author Response
Many thanks for your valuable time and constructive comments on our manuscript. Changes in the initial version of the manuscript are highlighted within the document using the track changes mode in MS Word. Below, we provide a point-by-point response to your comments.
- “In writing the abstract, I suggest that authors follow the prescribed instructions and add a background: Place the issue being addressed in a broad context and highlight the purpose of the study. Also, clarify the part that refers to the conclusion, as the last sentence is too general in this form.”
Response: Your suggestion is very imaginative and practical. We We have rewritten the abstract section of the paper, adding descriptions of the research background, research findings, and research significance.
- “The introduction clearly introduces the reader to the topic, citing the relevant literature and the uniqueness of this work compared to existing works. I suggest briefly describing the structure of the paper at the end of the introduction to give the reader an insight into the text of the following paper.”
Response: Thank you for your improvement suggestion. We have added the introduces of the article structure in the end of the introduction (lines 6-10 on page 3).
- “Lines 105-112: Since the chapter refers to previous work, the facts mentioned should be supported by literature. If it is the authors' opinion, it should be moved to the discussion chapter.”
Response: Thanks for this improvement suggestion. We have added related references to this chapter (lines 14-17 on page 3).
- “Line 113 starts with the term "Related research mostly starts..." and it is important to indicate which literature this part of the text refers to. This way it is easier to review and gain insight into the topic being addressed. Within the sentence, the authors only give the reference [28], and what is written suggests more than one literature.”
Response: Many thanks for your comments. We have clarified the reference of ‘related research’, and added some releated literatures (lines 22-24 on page 3).
- “The sentence in line 118 begins with the text "Few studies...", and again it is necessary to indicate which literature the facts mentioned refer to.”
Response: Thanks for giving us so valuable advice. In this context, we are summarizing the shortcomings of current research; therefore, we have made a correction to the sentence in line 27-29 on page 3.
- “The literature citation in line 136 contains an error: "Dargay & Clark [41]2012)".”
Response: Many thanks for your comments. We have checked all the manuscript and corrected all the ciation format.
- “Due to the extensive nature of the paper, it is necessary to depict the research process with a flowchart.”
Response: We agree with your comments. We have added a flowchart (Figure 3) to depict the research process on page 7.
- “"Error!Reference source not found." appears in many places in the paper, e.g., in line 258
Lines 287-88 should be deleted.”
Response: Thank you for pointing this out. After we did our best to read through the article, we fixed all the format error you pointed out.
- “Line 414, please check if "8‰ of leisure duration" is written correctly.”
Response: Many thanks for your comments. I have revised the wording of this sentence to indicate that this subset of activities may represent a very small number of leisure activities (8‰) that have been mistakenly identified (refer to lines 12-15 on page 14).
Additionally, after we did our best to read through the article, we fixed the readability issues in the manuscript. We have enlisted the assistance of a native English speaker to meticulously check the language and assist us in polishing it.
Round 2
Reviewer 2 Report
Comments and Suggestions for Authors I have reviewed the author changes to my review and the revised manuscript.All comments were responded to and corresponding changes were made. Comments on the Quality of English Language
Minor editing of English language required